# Ganglioside GD2 Enhances the Malignant Phenotypes of Melanoma Cells by Cooperating with Integrins

**DOI:** 10.3390/ijms23010423

**Published:** 2021-12-31

**Authors:** Farhana Yesmin, Robiul H. Bhuiyan, Yuhsuke Ohmi, Satoko Yamamoto, Kei Kaneko, Yuki Ohkawa, Pu Zhang, Kazunori Hamamura, Nai-Kong V. Cheung, Norihiro Kotani, Koichi Honke, Tetsuya Okajima, Mariko Kambe, Orie Tajima, Keiko Furukawa, Koichi Furukawa

**Affiliations:** 1Department of Biomedical Sciences, Chubu University College of Life and Health Sciences, Kasugai 487-8501, Japan; farhana7779@gmail.com (F.Y.); biochemistrobi79@gmail.com (R.H.B.); satoyama@isc.chubu.ac.jp (S.Y.); ikekoneka@isc.chubu.ac.jp (K.K.); yuki34@mc.pref.osaka.jp (Y.O.); apor0825@yahoo.co.jp (P.Z.); kambem@isc.chubu.ac.jp (M.K.); oriet@isc.chubu.ac.jp (O.T.); keikofu@isc.chubu.ac.jp (K.F.); 2Department of Molecular Biochemistry, Nagoya University Graduate School of Medicine, Nagoya 466-0065, Japan; tokajima@med.nagoya-u.ac.jp; 3Department of Medical Technology, Chubu University College of Life and Health Sciences, Kasugai 487-8501, Japan; ooumi82@isc.chubu.ac.jp; 4Department of Glyco-Oncology and Medical Biochemistry, Osaka International Cancer Institute, Osaka 541-8567, Japan; 5Department of Pharmacology, Aichi Gakuin University School of Dentistry, Nagoya 464-8650, Japan; hamak@dpc.agu.ac.jp; 6Memorial Sloan-Kettering Cancer Center, New York, NY 10065, USA; cheungn@mskcc.org; 7Department of Biochemistry, Saitama Medical University, Saitama 350-0495, Japan; kotani@saitama-med.ac.jp; 8Department of Biochemistry, Kochi University School of Medicine, Nangoku 783-8505, Japan; khonke@kochi-u.ac.jp

**Keywords:** ganglioside, cancer-associated antigen, integrin, GEM/rafts, melanoma

## Abstract

Gangliosides have been considered to modulate cell signals in the microdomain of the cell membrane, lipid/rafts, or glycolipid-enriched microdomain/rafts (GEM/rafts). In particular, cancer-associated gangliosides were reported to enhance the malignant properties of cancer cells. In fact, GD2-positive (GD2+) cells showed increased proliferation, invasion, and adhesion, compared with GD2-negative (GD2−) cells. However, the precise mechanisms by which gangliosides regulate cell signaling in GEM/rafts are not well understood. In order to analyze the roles of ganglioside GD2 in the malignant properties of melanoma cells, we searched for GD2-associating molecules on the cell membrane using the enzyme-mediated activation of radical sources combined with mass spectrometry, and integrin β1 was identified as a representative GD2-associating molecule. Then, we showed the physical association of GD2 and integrin β1 by immunoprecipitation/immunoblotting. Close localization was also shown by immuno-cytostaining and the proximity ligation assay. During cell adhesion, GD2+ cells showed multiple phospho-tyrosine bands, i.e., the epithelial growth factor receptor and focal adhesion kinase. The knockdown of integrin β1 revealed that the increased malignant phenotypes in GD2+ cells were clearly cancelled. Furthermore, the phosphor-tyrosine bands detected during the adhesion of GD2+ cells almost completely disappeared after the knockdown of integrin β1. Finally, immunoblotting to examine the intracellular distribution of integrins during cell adhesion revealed that large amounts of integrin β1 were localized in GEM/raft fractions in GD2+ cells before and just after cell adhesion, with the majority being localized in the non-raft fractions in GD2− cells. All these results suggest that GD2 and integrin β1 cooperate in GEM/rafts, leading to enhanced malignant phenotypes of melanomas.

## 1. Introduction

Gangliosides are sialic acid-containing glycosphingolipids, and they are expressed in almost all the cells and tissues of vertebrates [1]. In particular, complex gangliosides are commonly enriched in the nervous tissues of many animals in common, and have been considered to play important roles in the development and function of the nervous system [2]. On the other hand, some gangliosides were reported to be expressed in particular cancer cells and tissues, and so they have been considered to be cancer-associated carbohydrate antigens [3,4]. Among them, the gangliosides GD3 and GD2 have been used as markers for neuroectoderm-derived cancers, and also as targets of immunotherapy, such as antibody therapy [5,6,7].

Since the cDNAs of ganglioside synthetic enzymes were isolated, it became possible to investigate the roles of gangliosides in various cells and tissues [8]. In particular, the genetic engineering of glycosyltransferase genes in cultured cells and experimental animals have enabled us to clarify significant roles of gangliosides, and their mechanisms in development and carcinogenesis [9]. 

Although it became possible to compare the phenotypic changes of glyco-remodeling cells and animals, the mechanisms by which gangliosides modulate the phenotypes and cell signals have remained unclear. This is because glycosphingolipids are expressed on the outer layer of the lipid bilayer membrane [10], and it can be difficult to mediate cell signals that are introduced via the cell membrane. The novel approach of EMARS/MS (enzyme-mediated activation of radical sources/mass spectrometry) has led to a breakthrough in this issue. EMARS/MS was developed by Kotani and Honke [11], and has been verified to be a powerful method to identify interacting molecules with some target antigens on the cell surface [12]. Since we use living cells to analyze events on the cell surface, corresponding to the size of membrane microdomains, this method uses no special equipment and is applicable for a comprehensive analysis of clustering molecules with particular targets [11]. We have reported the interesting molecular associations of gangliosides with newly-defined membrane molecules in melanomas [13] and gliomas [14]. Thus, the functional analysis of cancer-associated glycolipids is entering a new era [15].

Among the cancer-associated glycolipids, GD2 is specifically important because of its key roles in the metastasis of melanomas [16], as a marker of cancer stem cells for breast cancers [17] and triple-negative breast cancers [18], and as targets of novel immune therapy for neuroectoderm-derived cancers [19] and other cancers, too [20].

In this study, we identify the membrane molecules interacting with ganglioside GD2 on the surface of human melanoma cells using EMARS/MS, and integrins were identified as representative molecules to associate with GD2. Furthermore, not only is there a close connection between GD2 and integrins, but the study also elucidated their marked cooperation in the augmentation of cancer phenotypes, particularly in cell adhesion, proliferation, and invasion.

## 2. Results

### 2.1. Establishment and Confirmation of GD2+ Melanoma Cell Lines

Using a subline of the human melanoma cell line SK-MEL-28 (N1) [21], GD2-positive lines, S6 and S2, were established based on the synthetic pathway (Figure 1A), as previously described [22]. We selected strong GD2-expressing, but not GD3-expressing, lines in order to clearly identify the specific function of GD2. The expression pattern of GD2 and GD3 is shown in Figure 1B. The detection of GD2 by mAb 3F8 is shown in Appendix A. Negative control lines, V4 and V9, were also established, which were neo-resistant but not expressing GD2 or GD3. The expression of GD2 was confirmed by immunocytochemistry, as shown in Figure 1C.

### 2.2. GD2 Expression Resulted in the Increased Malignant Properties of Melanomas 

Cell proliferation was analyzed using GD2+ and GD2− cells by MTT. As shown in Figure 2A, S1 and S6 showed a higher proliferation activity than V4 and V9 with significance (* *p* < 0.05). When the invasion activity was compared between GD2+ and GD2− cells, the GD2+ cells showed greater invasion than GD2− cells (Figure 2B). In this experiment, the upper chamber of inserts was coated with Matrigel, and FCS was added to the lower chamber. Giemsa-stained cells on the reverse side of the membrane are shown in Figure 2C. In order to examine the cell adhesion activity, the RT-CES system was used. After plating cells in collagen I-precoated microplates, the cell index was monitored for 24 h. The GD2+ cells showed stronger adhesion activity than the GD2− cells (Figure 2D). 

### 2.3. Anti-GD2 Monoclonal Antibody Suppressed the Increased Malignant Properties of GD2+ Cells

In order to clarify the involvement of GD2 in the malignant properties, anti-GD2 monoclonal antibodies (mAbs) were added during the measurement of cell growth and adhesion. To examine the effects of anti-GD2 mAb on cell growth, the MTT assay was performed using GD2+ S1 and GD2− V4 cells. Purified anti-GD2 mAbs 220-51 were added to each well at 72, 18, and 4.5 μg/mL for days 0–5. Anti-GD2 mAb-treated GD2+ cells showed significant growth suppression with anti-GD2 mAb at 72 and 18 μg/mL, compared with non-treated cells (Figure 3A, left). However, GD2− cells did not show significant differences between treated and non-treated cells (Figure 3A, right).

The effects of treatment with anti-GD2 mAb 220-51 on cell adhesion were analyzed by the RT-CES system using GD2+ and GD2− cells in collagen I-precoated microplates. When anti-GD2 Ab 220-51 was added to GD2+ cells at 0.5 h (Figure 3B, left) after starting, cell adhesion was markedly suppressed for ~12 h, but gradually recovered thereafter until 24 h. When the antibody was added at or 3.0 h (Figure 3C, left) after starting, cell adhesion was stably suppressed from 0.5 h to 24 h during the observation. On the other hand, when GD2− cells were treated by anti-GD2 mAb, there were no significant differences between the mAb-treated and -non-treated cells (Figure 3A,B, right). Treatment with purified anti-GD2 mAb 3F8 showed essentially similar results for GD2+ and GD2− cells, respectively (Appendix A, left and right). Thus, anti-GD2 mAbs strongly suppressed cell adhesion only for GD2+ cells.

### 2.4. Identification of Integrin Β1 as A GD2−Associating Molecule with EMARS/MS

EMARS (Figure 4A) was performed with GD2+ S1 cells using purified mouse anti-GD2 mAb 220-51. The FITC-labeled molecules were immunoprecipitated with a rabbit anti-FITC antibody, and detected by immunoblotting with a goat anti-FITC antibody, as shown in Figure 4B. In the results of the MS analysis, there were more than 30 molecules detected, as listed in Appendix A. Among them, integrin β1 was defined as a cell surface molecule detected only in mAb-treated cells, possibly associated with GD2. Among the 13 molecules defined, only integrin β1 was a definite membrane molecule. Detailed procedures for LC-MS are described in the Appendix A. The detection of integrin β1 by MS is also summarized in Appendix A.

Cell surface expression and mRNA expression levels of integrin β1 in GD2+ and GD2− cells were analyzed by flow cytometry using anti-integrin β1 mAb and RT-qPCR, respectively (Figure 5A,B). Both the surface expression and mRNA of integrin β1 showed almost equivalent levels between GD2+ and GD2− cells. The expression of integrin β1 as well as GD2 was analyzed by immunoblotting, showing similar protein levels of integrin β1 in all samples (Figure 5(Ca)). The binding of ganglioside GD2 and integrin β1 was analyzed by immunoprecipitation and subsequent immunoblotting using GD2+ and GD2− clones. Cell lysates from GD2+ and GD2− cells were used for immunoprecipitation with rabbit anti-integrin β1 antibodies, and the immunoprecipites were immunoblotted separately with mouse anti-GD2 mAb or mouse anti-integrin β1 mAb. We observed GD2+ (S1 and S6) cells and integrin β1 in the same lanes, indicating that integrin β1 and GD2 were associated on the cell membrane (Figure 5(Cb)).

### 2.5. Colocalization and Close Association of GD2 and Integrins as Shown in Image Analyses 

Colocalization of GD2 and integrin β1 on the cell surface was examined by immunocytochemistry, using GD2+ S1 and S6 cells (Figure 6A). The majority of GD2 (red) was colocalized with integrin β1 (Figure 6A, right). The interaction between GD2 and integrin β1 was analyzed by PLA. As shown in Figure 6B, after fixation, GD2+ and GD2− cells were incubated with anti-GD2 and anti-integrin β1 mAbs. Then, Duolink in situ PLA probes anti-mouse PLUS and anti-rabbit MINUS were added and incubated. After the addition of a ligation–ligase solution, an amplification reaction was carried out. Under a confocal microscope, amplification products were detected as red membrane staining patterns in GD2+ cells (Figure 6B). These results suggest that GD2 and integrins cluster on the cell surface.

### 2.6. Increased Cell Signals in GD2+ Cells as Shown by Tyrosine-Phosphorylated Proteins during Cell Adhesion to Collagen I

Cell signals were analyzed by immunoblotting with anti-phosphotyrosine mAb PY20. As shown in the diagram in Figure 7A, cells in the plain DMEM were rotated for 1 h after trypsinization, and added to collagen I-precoated plates. After incubation, as indicated, the cell lysates were analyzed by immunoblotting. As shown in Figure 7B, mainly 3 bands at 180, 130, and 100 kDa were strongly detected in GD2+ cell-derived samples. The identification of these bands was performed by MS, revealing them to be EGFR and FAK (upper two bands). Detailed data of MS analysis to define these molecules are shown in Appendix A.

### 2.7. Knockdown of Integrin Β1 in GD2+ Cells Altered Their Phenotypes to Those of GD2− Cells

Effects of 4 siRNAs to compare the knockdown efficiency of integrin β1 (37, 75, 74, and ITG1) were examined with Western immunoblotting (Figure 8A) and qRT-PCR (Figure 8B). Then, we selected ITG1 si-RNA and examined the effects of the knockdown of integrin β1 on cell proliferation. As shown in Figure 8C, GD2+ cells show a clear reduction of cell growth after knockdown with ITG1 compared with GD2+ control cells, but GD2− cells do not show a significant difference from the levels of GD2− control cells. As for cell adhesion activity, the treatment with siRNA results in a significant decrease in adhesion, as shown in GD2+ cells (Figure 8D). The reduction of integrin β1 was also shown. Regarding the effects of the knockdown on the invasion activity, the increased invasion activity of GD2+ cells was strongly suppressed to the levels of GD2− cells (Figure 8F), and their images are shown in Figure 8E. 

Immunoblotting with anti-phosphotyrosine mAb PY20 was performed using lysates from cells transfected with anti-integrin β1 si-RNA ITG1. After a 36 h culture in a regular medium, and the subsequent starvation in serum-free DMEM for 23 h, cells were incubated in collagen I precoated plates for 0~30 min, and underwent immunoblotting with PY20. Surprisingly, the phosphorylated bands in GD2+ cell-derived samples largely disappeared, as shown in Figure 9. All these results suggest that the increased malignant properties and signals detected in GD2+ cells are dependent on the expression of integrins, based on their association and cooperation on the cell surface.

### 2.8. Integrin Β1 and GD2 Co-Localized in GEM/Rafts before and during the Adhesion to CL Type I in GD2+ Cells

Cells were detached using 0.02% EDTA/PBS, and placed in collagen I pre-coated plates in DMEM. After incubation for 0~30 min at 37 °C, cell lysates were prepared and fractionated by Optiprep gradient ultracentrifugation at 42,000 rpm and 4 °C for 5 h. These fractions were used for immunoblotting with anti-integrin β1 mAb, anti-GD2 mAb, anti-flotillin, and anti-caveolin-1 antibodies, as shown in Figure 10A. Band intensities of integrin β1 were plotted. As shown in Figure 10B, integrin β1 was found in fractions (fr.) 2~4 as well as in fr. 6~10 in GD2+ cells. On the other hand, integrin β1 can only be found in fr. 6~10 in GD2− cells at 0~5 min, during adhesion. After 15 min, integrin β1 was broadly distributed in both the GD2+ and GD2− cells. 

## 3. Discussion

Gangliosides are expressed at high levels in the nervous tissues among normal tissues [1]. However, their characteristic expression in neuroectoderm-derived malignant cells, such as malignant melanomas, neuroblastomas, gliomas, and small cell lung cancers, [4] was reported, leading to their expected clinical application as cancer-associated glycolipids. They have also been reported to be expressed in osteosarcomas [23,24,25], breast cancers [26], and T-cell leukemias [27,28,29,30]. Attention to GD2 is especially increasing, i.e., as cancer-associated glycolipids in various cancers, and a marker to indicate higher malignant properties of cancers [22], and/or cancer stem cells [17] and triple-negative breast cancers [18]. GD2 has been used as a target of antibody therapy [31,32] towards neuroblastomas [33], gliomas [34], and breast cancers [26], and also of CAR-T therapy of various cancers [19,20]. Furthermore, the biological function of GD2 has been reported, i.e., in EMT [35,36], tumor invasion [37], and cancer metastasis [38], though it had been expected based on the clinical samples with different disease stages [16]. Thus, they have attracted interest from medical and biological science fields. In particular, GD2 is now expected to be a target to protect cancer metastasis, since metastasis is the most serious cause for cancer death. However, it is not well-understood how GD2 is involved in cancer metastasis. 

EMARS/MS was developed by Kotani and Honke [11], and has been shown to be an efficient approach to investigate the mechanisms by which membrane molecules exert significant effects on the cell membrane [15]. Particularly for glycosphingolipids, the identification of physically associating molecules on the cell surface can be crucial to understand their roles in cell signal transduction [39], because of their modes of membrane anchoring. Glycosphingolipids are expressed on the outer layer of the lipid bilayer membrane lacking a cytoplasmic domain [40]. Therefore, they need to physically associate with membrane molecules containing cytoplasmic domains, in order to modulate signals introduced via the cell membrane. Then, the EMARS/MS approach should bring about enormous progress in the understanding of the mechanisms for signal regulation by glycosphingolipids. Thus, the functional analysis of cancer-associated glycolipids is now entering a new era [15].

The current biochemical isolation method of lipid rafts contains some unclear issues, i.e., the isolated raft fraction members do not necessarily co-exist on the same membrane microdomains. On the other hand, molecules defined by EMARS/MS with a particular target can directly be considered to be present closely enough to be expected to physically and functionally cooperate on the living cells. 

In this study, we identified integrin β1 as a representative membrane molecule in GD2+ melanoma cells. Previously, we reported that neogenin is a GD3-associated membrane molecule on melanoma cells [13], and elucidated that the intracellular domain of neogenin was cleaved by γ-secretase in GEM/rafts, and plays important roles as a GD3-drived effector [13]. It is very interesting that the EMARS-MS analysis revealed different membrane molecules associating with two gangliosides, GD3 and GD2, although these ganglioside-expressing transfectant cells were generated from the same melanoma cell, an SK-MEL-28 mutant, N1 cell line [16]. These results suggest that individual gangliosides form distinct molecular clusters on the cell membrane, and play individually characteristic roles in the processes of cancer generation, progress, and expansion.

Molecular clustering around GD2 in GEM/rafts should be further clarified, and the association of GD2/integrins with EGFR and FAK, for instance, is also an area to be urgently examined. This is because these molecular clusters, formed around cancer-associated gangliosides, can be promising targets of cancer treatment with higher specificity and stronger efficiency.

While close relationships between gangliosides and integrins have been reported [41], we also showed that ganglioside GD3 enhanced the cell adhesion of melanoma cells by forming a molecular complex with integrins [42], and the association of GD2 and integrins can be more intense when considering its role in cancer metastasis. It is surprising that the increased malignant properties based on GD2 expression were cancelled by the knockdown of integrin β1, suggesting that GD2 is essentially involved in cell adhesion, depending on integrin β1. In fact, anti-GD2 mAb very strongly suppressed cell adhesion, just as observed by the knockdown of integrin β1, as shown in Figure 3.

Whether GD2 is always positively involved in cell adhesion is unclear. At this moment, it is considered to be dependent on the types of cancers. In RT-CES, GD2+ melanoma cells showed a stronger cell adhesion in this melanoma study. On the other hand, GD2 expression resulted in the apparent reduction of cell adhesion in osteosarcoma cells, as previously reported [23]. This discrepancy in the effects of GD2 expression on the cell adhesion depends on the type of cancer cells, i.e., melanomas and osteosarcomas in which different signaling molecules are expressed [43]. As for cell invasion activity, GD2+ cells showed increased invasion activity, compared with GD2− cells in this study. In a previous report, we observed a reduced invasion activity in GD2+ melanomas, as measured by the Boyden chamber method [22]. The difference in the invasion activity of GD2+ cells should come from different experimental settings, i.e., the presence or absence of FCS in the lower chamber that acts as a chemoattractant. Thus, GD2+ cells can show rather reduced invasion under the no FCS condition. These results show that we must carefully select experimental conditions for cell phenotypic analyses. 

Consequently, we need to elucidate the effects of cancer-associated gangliosides to distinguish the universal effects and specific effects in some particular cancers on an individual basis. Then, we will be able to apply anti-ganglioside probes to treat individual cancer types.

## 4. Materials and Methods

### 4.1. Cell Culture

Ganglioside GD2−expressing S1 and S6 clones were established from a human melanoma cell SK-MEL-28 subline, N1 [21], by transfecting with cDNAs of *ST8SIA1* [44] and *B4GALNT1* [45]. The N1 cell was provided by K.O. Lloyd (Memorial Sloan Kettering Cancer Center, New York, NY, USA), as reported in [21]. V4 and V9 clones were established as vector controls [22] (Figure 1A). S1 and S6 clones were grown in Dulbecco’s minimal essential medium (DMEM) containing 7.5% fetal bovine serum (FBS) and G418 (600 μg/mL), and vector control V4 and V9 clones were grown in DMEM containing 7.5% FBS and G418 (400 μg/mL) at 37 °C in a humidified atmosphere containing 5% CO_2_.

### 4.2. Mice

Eight-week-old immunocompromised mice (Chubu Science Co. Ltd., Nagoya, Japan) were used to generate ascites of anti-GD2 mAb 220-51. Animal experimental protocols were approved by the committee on Laboratory Animals in Chubu University (No. 3010006), consistent with the guidelines of the Japanese government, as well as the National Institutes of Health Guide for the Care and Use of Laboratory Animals (1966). Mice were maintained under specific pathogen-free conditions. 

### 4.3. Antibodies and Reagents 

Anti-GD2 mAb, 3F8, was provided by N. K. Cheung (Sloan Kettering Cancer Center, New York, NY, USA) [46] and 220-51 was generated in our laboratory [47]. Anti-GD3 mAb and anti-GM2 mAbs were provided by L.J. Old at the Memorial Sloan Kettering Cancer Center (New York). The other antibodies and reagents were obtained from the following commercial sources: fluorescein isothiocyanate (FITC)-conjugated goat anti-mouse IgG (H + L) from Cappel (Durham, NC, USA); horseradish peroxidase (HRP)-conjugated anti-mouse IgG antibody; HRP-conjugated anti-rabbit IgG antibody; HRP-conjugated anti-goat IgG antibody; rabbit anti-FITC antibody; and goat anti-FITC antibody from Cell Signaling Technology (Danvers, MA, USA). Mouse anti-integrin β1 mAb, anti-Flotillin-1 mAb, anti-Caveolin-1 Ab, and anti-integrin β1(CD29) mAb 4B7R were from Santa Cruz Biotechnology (Santa Cruz, CA, USA). Rabbit anti-integrin β1 polyclonal Ab was from Bioss, Woburn, MA, USA. Mouse anti-phosphotyrosine antibody PY20 was from Santa Cruz Biotechnology (Santa Cruz, CA, USA). Alexa 488-conjugated anti-rabbit IgG antibody and Alexa 568-conjugated anti-mouse IgG antibody were from Invitrogen (Carlsbad, CA, USA). Monoclonal anti-β actin antibody was from Sigma-Aldrich (St. Louis, MO, USA). Protein A Sepharose^TM^ 4 Flast Flow beads were from GE Healthcare. Giemsa was from Wako (Osaka, Japan), Matrigel^TM^ was from BD Bioscience and collagen-1 was from Sigma-Aldrich (St. Louis, MO, USA). FITC-conjugated tyramine (FT), BSA, dithiothreitol, iodoacetamide, ammonium bicarbonate, and Lys-C were from Wako (Osaka, Japan). Cell lysis buffer was from Cell Signaling Technology (Danvers, MA, USA), and Protease Inhibitor MixtureTM was from Calbiochem (San Diego, CA, USA). The ImmunoStar^TM^ LD detection kit was from Wako (Osaka, Japan), and OptiPrep^TM^ and 1% 020 Brij were purchased from Sigma-Aldrich (St. Louis, MO, USA). 

### 4.4. Flow Cytometry

The expression levels of ganglioside GD2 on the cell membrane were analyzed by Accuri^TM^ C6 Flow Cytometer (Accuri Cytometers Inc., Ann Arbor, MI, USA), as previously described [48,49]. Briefly, trypsinized cells (5 × 10^5^) were washed twice with cold PBS, and incubated with diluted antibodies in PBS for 60 min on ice. After washing twice with PBS, cells were stained with FITC-conjugated goat anti-mouse IgG (H + L) (Cappel, Durham, NC, USA) as a secondary antibody for 45 min on ice. Then, cells were washed twice with PBS, and the relative expression levels were analyzed by flow cytometry. Control samples were prepared using non-relevant mAbs with the same subclasses as the individual primary antibodies. The CFlow plus^TM^ program was used for the quantification of positive cells.

### 4.5. Immunofluorescence Assay

Cells (500 cells/well/10 μL) were seeded in each well of 60-well Terasaki plates, and incubated in regular medium overnight. The supernatants were removed gently, and the diluted antibodies were applied to individual wells and incubated for 1 h at room temperature (RT). After being washed twice with PBS containing 5% FBS, FITC-conjugated anti-mouse IgG (H + L) (Cappel, Durham, NC, USA) was applied, and the plates were incubated for 1 h at RT. Then, the cells were washed twice with PBS containing 5% FBS. Antigen was detected by fluorescence microscopy (IX2-ILL100, OLYMPUS, Tokyo, Japan).

### 4.6. MTT Assay

Cells (3 × 10^3^) were seeded in each well of 96-well plates with 100 μL of DMEM supplemented with 7.5% FBS. During culture, 10 μL MTT (3-(4,5-dimethylthiazol-2-yl)-2,5-diphenyltetrazolium bromide) (5 mg/mL PBS) solution was added to each well on days 0, 1, 2, 3, 4, 5, 6, 7, 8, and 9, and incubated for 4 h at 37 °C. The reaction was stopped by adding 110 μL of 1-propanol containing 0.4% HCl and 0.1% NP-40. Then, absorbance was measured at 590 nm using an automatic microplate reader (Thermo Fisher Scientific, Type: 357, Shanghai, China).

### 4.7. Invasion Assay

An invasion assay was performed using cell culture inserts (Transparent PET^TM^ membrane, 24-well format, 8.0-μm pore size, Life Sciences, Durham, NC, USA), as described [14]. Matrigel^TM^ (BD Bioscience) 20 μL in cold PBS (200 μg/ mL) was added to the upper chamber of cell culture inserts, and incubated for 2 h at room temperature to promote polymerization. After removing PBS, the upper chamber was filled with 200 μL serum-free DMEM and incubated for 1 h, and the lower chamber was filled with DMEM containing 10% FBS. After removing the medium from the upper chamber, cells (3 × 10^4^) in 200 μL serum-free DMEM were added, and incubated for 24 h at 37 °C in a humidified atmosphere containing 5% CO_2_. After incubation, invaded cells on the surface of the lower chamber were stained with Giemsa (Wako, Osaka, Japan), and the number of cells was counted under microscope (IX73P1F^TM^, Olympus, Tokyo, Japan).

### 4.8. Cell Adhesion Assay

The cell adhesion experiment was performed using the real-time cell electronic sensing system (RT-CES^TM^) (Wako Pure Chemical, Osaka, Japan), as previously described [22]. At the bottom of the microplates (E-Plate (16X) (ACEA Biosciences Inc., San Diego, CA, USA), microelectronic cell sensor arrays are integrated. The sensor provides information on the increased electrical resistance (cell index), indicating the increase in cell adhesion. E-plates were coated with Collagen-1 (CL-1) (5 μg/m in PBS, 100 μL/well) at RT for 1 h, and blocked by 1% BSA/10% FBS in D-MEM (100 μL/well) at RT for 1 h. After blocking the wells, the cells (1 × 10^4^) were seeded in each well of the plates containing the culture medium. Changes of cell adhesion were monitored continuously, and expressed as the cell index. 

### 4.9. Cell Lysate Preparation

Cells were washed three times with PBS, and lysed using a cell lysis buffer (20 mM Tris-HCl, 150 mM NaCl, 1 mM EGTA, 1 mM Na2EDTA, 1% Triton X-100, 2.5 mM sodium pyrophosphate, 1 mM β-glycerophosphate, 1 mM Na3VO4, and 1 µg/mL leupeptin) (Cell Signaling Technology, Danvers, MA, USA), supplemented with 1 mM phenylmethylsulfonyl fluoride (PMSF) and Protease Inhibitor MixtureTM (Calbiochem, San Diego, CA, USA). Cell lysates were centrifuged at 12,500 rpm (Kubota 3740TM, Tokyo, Japan) for 10 min at 4 °C to remove insoluble cell debris, and proteins in supernatants were measured using the DC protein assay kit (Bio-Rad, Hercules, CA, USA).

### 4.10. EMARS and MS Analysis

The EMARS reaction and MS of EMARS products were carried out, as described [11,50,51]. Briefly, the cells (5.0 × 10^5^) were treated with purified mouse anti-GD2 mAb 220-51 in 6-cm dishes after washing with PBS, and incubated for 45 min at RT. Then, HRP-conjugated anti-mouse IgG in PBS was added and incubated for 30 min at RT. After washing twice with PBS, (FITC)-conjugated tyramine (FT) solution diluted in PBS was added (0.1 mM FT/PBS) to the dishes and incubated in dark place for 15 min at RT. After the addition of H_2_O_2_ to the dishes and 5 min incubation, cells were washed twice with PBS and scraped by adding 100 mM Tris-HCl (pH 7.4) and 1 mM PMSF. After centrifugation, pellets were dissolved by adding an RIPA buffer (50 mM Tris HCl, pH 7.4, 150 mM NaCl, 1% NP40. 0.5% sodium deoxycholate, and 0.1% SDS), and 1 mM phenylmethylsulfonyl fluoride (PMSF) was then added. The FITC-labeled molecules were immunoprecipitated with the rabbit anti-FITC antibody, and the efficiency of the EMARS reaction was confirmed by SDS-PAGE and the subsequent immunoblotting using goat anti-FITC antibody.

Then, immunoprecipitates were prepared for MS analysis by dissolving with an MS sample buffer (12 mmol/L sodium lauroylsarcosine, 12 mmol/L sodium deoxycholate, and 100 mmol/L Tris-HCl (pH 8.0)), then boiling at 95 °C, and centrifugation at 20,000 g for 15 min. Dithiothreitol and iodoacetamide were used for reduction and alkylation, respectively. Then, the samples were diluted with 50 mmol/L ammonium bicarbonate and digested by Lys-C (Wako, Osaka, Japan) for 3 h, followed by digestion with trypsin for 8 h at 37 °C. The samples were desalted and concentrated with C18 Stage-Tips™ (Thermo Fisher Scientific, Waltham, MA, USA). Mass spectrometry was performed using an LTQ-Orbitrap Velos mass spectrometer (Thermo Fisher Scientific) system coupled with nano-LC (EASY-nLC II, Thermo Scientific). Detailed procedures were described in the Supportive information. Combined MS spectra (Tandem) were processed using Proteome Discoverer software (version 1.3, Thermo Fisher Scientific) workflow and searched using the program Mascot 2.4 (Matrix Science, Boston, MA, USA) and X! Tandem (The Global Proteome Machine; http://www.thegpm.org/tandem/), against the Swiss-Prot protein database. This was performed on 12 November 2018.

As for the identification of tyrosine-phosphorylated bands detected by the PY20 antibody in immunoblotting, an MS analysis of tyrosine-phosphorylated proteins was performed by using gel extracts, as previously described [49]. 

### 4.11. Western Blotting 

After preparation of the cell lysates, the proteins in cell lysates were separated in SDS-PAGE using 10% gels. The separated proteins in gels were transferred onto an Immobilon-P membrane (EMD Millipore, Burlington, MA, USA), and blots were blocked for 1 h with 3% skimmed milk or bovine serum albumin (BSA) in PBS, including 0.05% Tween-20. Then, the membrane was incubated with primary antibodies, followed by HRP-labeled secondary antibodies. Bands of proteins were visualized using ImmunoStar^TM^ LD detection kits (Wako Osaka, Japan), and analyzed by Imager LI-COR^TM^ (Model: 3600, LI-COR, Inc., Lincoln, NE, USA).

### 4.12. Immunoprecipitation

Cells were washed three times with PBS and lysed to prepare cell lysates, as mentioned above. After removing insoluble cell debris through centrifugation, The supernatants of cell lysates were used for the immunoprecipitation of integrin β1 with rabbit anti-integrin β1 polyclonal Ab (Bioss, MA, USA), or the immunoprecipitation of phosphorylated signaling molecules with the mouse anti-phosphotyrosine antibody PY20 (Santa Cruz Biotechnology, Santa Cruz, CA, USA) at 4 °C overnight with rotation. Protein G Sepharose 4 fast flow^TM^ beads (GE Healthcare, Uppsala, Sweden) were used to capture the immune complex. After washing the beads with immune complex, a sodium dodecyl sulfate (SDS) sample buffer (2X) was added and boiled at 95 °C for 3 min. Then, an immunoprecipitated complex was applied for SDS–polyacrylamide gel electrophoresis (SDS–PAGE).

### 4.13. Real-Time RT-PCR

Total RNA extraction was performed with a TRIzol^TM^ reagent (Invirogen), and a cDNA template was synthesized from the total RNA using the MMLV reverse transcriptase kit (Invitrogen), as described previously [49]. Briefly, 2 µg total RNA was used for the synthesis of cDNA with MMLV reverse transcriptase. Synthesized cDNA (8 ng) was amplified in a 20 µL reaction volume containing a 10 µL SsoAdvanced^TM^ Universal SYBR green Supermix^TM^ qPCR kit (Bio-Rad Laboratories, Hercules, CA, USA) and 1 µL of each 5 µM primer. The primers were designed as forward (5′-GCTGGTGTGGTTGCTGGAATTG-3′) and reverse (5′- GACCACAGTTGTTA-CGGCACTC-3′), for integrin β1. The PCR program was carried out with initial denaturation at 95 °C for 30 s, followed by 40 cycles of amplification (95 °C for 5 s, 58 °C for 30 s, and 65 °C for 5 s). 

### 4.14. Immunocytochemistry

Immunocytochemistry was performed, as previously mentioned [52]. GD2+ (S1 and S6) and GD2− (V4 and V9) cells were plated in glass-bottomed dishes (Iwaki, Tokyo, Japan) and incubated in DMEM containing 7.5% FBS at 37 °C for 24 h. After washing with cold PBS, the cells were fixed with 4% paraformaldehyde in PBS for 10 min at room temperature (RT). Cells were blocked with 5% BSA in PBS for 1 h at RT. Immunostaining was performed with anti-GD2 mAb (m-220-51) and anti-integrin β1 polyclonal Ab (Bioss, Massachusetts, USA) in 2% BSA/PBS as primary antibodies for 1 h at RT. Samples were washed with 1% BSA/PBS, and incubated with an Alexa fluor 568 conjugated goat anti-mouse-IgG (Invitrogen) and an Alexa fluor 488 conjugated donkey anti-rabbit IgG (Invitrogen) in 2% BSA/PBS for 1 h at RT. After being washed, the cell nucleus was stained with DAPI in 2% BSA/PBS for 15 min, and mounted with Pro-Long anti-fade reagent. Then, the cells were analyzed by a confocal microscope (Fluoview FV10i; Olympus, Tokyo, Japan).

### 4.15. Preparation of GEM/Rafts Fractions

Cells (2.5 × 10^7^) were washed with cold PBS and lysed with TNE buffer (25 mM Tris-HCL (pH 7.4), 150 mM NaCl, and 5 mM EDTA) containing BRIJ™020 (Sigma-Aldrich, St. Louis, MO, USA) and 1 mM PMSF. Lysates were homogenized 10 times with a Digital Homogenizer^TM^ (AS ONE, Osaka, Japan) or suspended with a 24-gauge needle. Homogenates were incubated on ice for 30 min, and centrifuged at 4 °C, 1000 rpm for 5 min to remove insoluble materials. The lysates (350 µL) were mixed with 700 µL OptiPrep^TM^ (Sigma-Aldrich, St. Louis, MO, USA) to a final concentration of 40% OptiPrep [53]. The mixture was carefully overlaid with 2 mL of 30% OptiPrep in TNE, followed by 950 µL of 5% OptiPrep. Gradient formation was performed by centrifugation for 5 h at 4 °C and 42,000 rpm using a Beckman MLS50 rotor (Kent, MI, USA). Each fraction was collected as 400 µL from the top of the gradient and applied for Western immunoblotting.

### 4.16. Proximity Ligation Assay (PLA)

The proximity ligation assay (PLA) was performed using a Duolink in situ PLA kit (Sigma-Aldrich), according to the manufacturer’s procedure. In brief, cells (2 × 10^5^) were plated in a glass base dish (Iwaki/Asahi Glass Co., Funahashi, Japan) pre-coated with 0.01% poly-L-lysine (Sigma-Aldrich), and incubated for 24 h in DMEM supplemented with 7.5% FBS. After washing with cold PBS, the cells were fixed with 4% paraformaldehyde in PBS for 10 min at RT. Then, the cells were blocked with 10% donkey serum in PBS for 1 h, and incubated with mouse anti-GD2 mAb (220-51) and rabbit anti-integrin β1 Ab in 0.5% donkey serum albumin in PBS for 1 h at RT. The cells were washed twice with 0.05% Tween-20, and incubated with the Duolink in situ PLA probes anti-mouse PLUS and anti-rabbit MINUS for 1 h at 37 °C. After washing twice with buffer A (0.01 M Tris, pH 7.4, 0.15 M NaCl, and 0.05% Tween 20), a ligation solution was added and incubated for 30 min at 37 °C. Amplification was carried out using amplification reagent-polymerase solution for over 100 min at 37 °C. The samples were dried for approximately 10 min at room temperature in the dark, and mounted with a minimal volume of ProLong Gold antifade reagent with DAPI. The cells were analyzed under a confocal microscope (Fluoview FV10; Olympus, Tokyo, Japan).

### 4.17. Statistical Analysis

Data are presented as the means ± SD. The data were analyzed by two-way ANOVA with the a Tukey post-hoc test or an unpaired Student’s two-tailed *t*-test, to compare mean values, as indicated in the individual figure legends. The *p*-values of <0.05 were considered significant. Those statistical significances were analyzed using R software (version 3.6.3) (https://www.R-project.org). The analysis was performed on 1 September 2020.

## Figures and Tables

**Figure 1 ijms-23-00423-f001:**
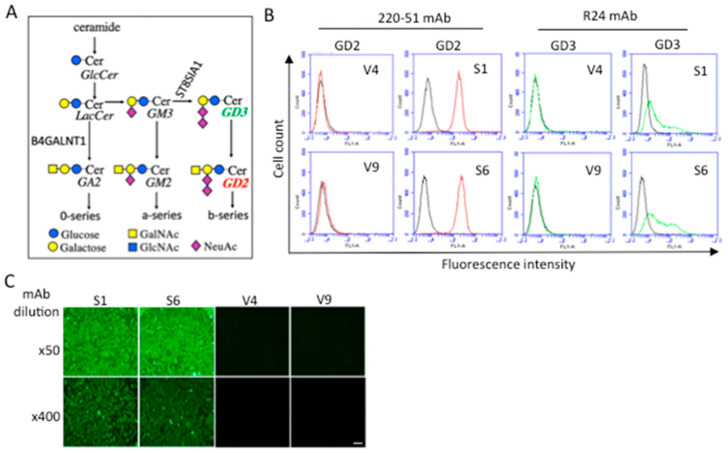
Establishment of cell lines expressing ganglioside GD2. (**A**) Biosynthetic pathway of gangliosides and their synthetic enzymes. (**B**) Expressions of gangliosides GD2 and GD3 in GD2+ (S1 and S6) and GD2− (V4 and V9) cells were analyzed by flow cytometry, using anti-GD2 and anti-GD3 mAbs. (**C**) Immunofluorescence images of the expression of GD2 in S1 and S6 cells using purified anti-GD2 mAb 220-51 at 4.5 and 36 μg/mL, and an FITC anti-mouse IgG 2nd antibody. Scale bar: 50 μm.

**Figure 2 ijms-23-00423-f002:**
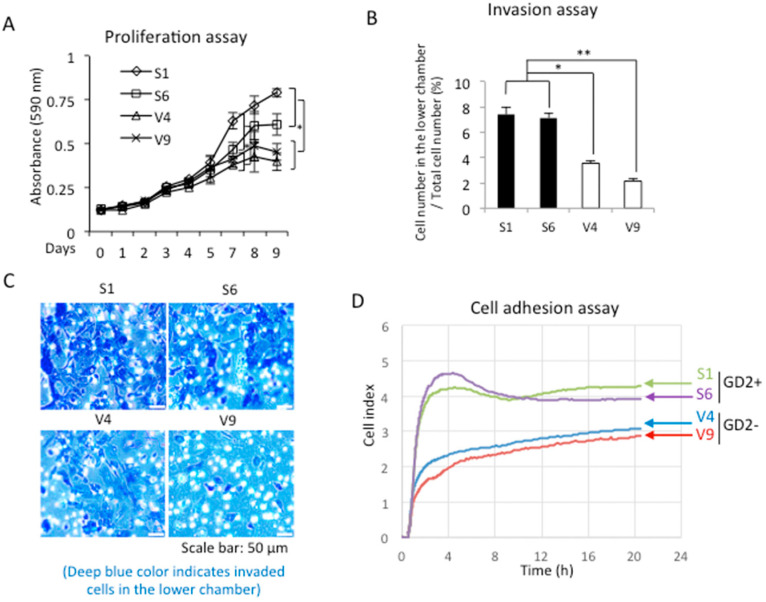
GD2+ cells showed increased proliferation, invasion, and cell adhesion. (**A**) Proliferation was analyzed using GD2+ and GD2− cells by the MTT assay. Cells were seeded and MTT solution was added and incubated on each day, as indicated. Absorbance was measured at 590/620 nm and relative absorbance was plotted. MTT assay was performed in triplicates. The means ± SD are presented, and analyzed using the two-way ANOVA with a Tukey post-hoc test between V-series vs. S-series cell lines. * *p* < 0.05. (**B**) Invasion activity was analyzed using GD2+ cells and GD2− cells with cell culture inserts. After 24 h of incubation, cells that invaded the lower chamber were stained with Giemsa, and the number was counted under a microscope. The invasion assay was performed in triplicates. The means ± SD are presented and analyzed using Student’s *t*-test. * *p* < 0.05 and ** *p* < 0.01. (**C**) Microscopic images of invaded cells are shown. Scale bar: 50 μm. (**D**), cell adhesion was analyzed by the RT-CES system using GD2+ and GD2− cells in collagen I-precoated plates.

**Figure 3 ijms-23-00423-f003:**
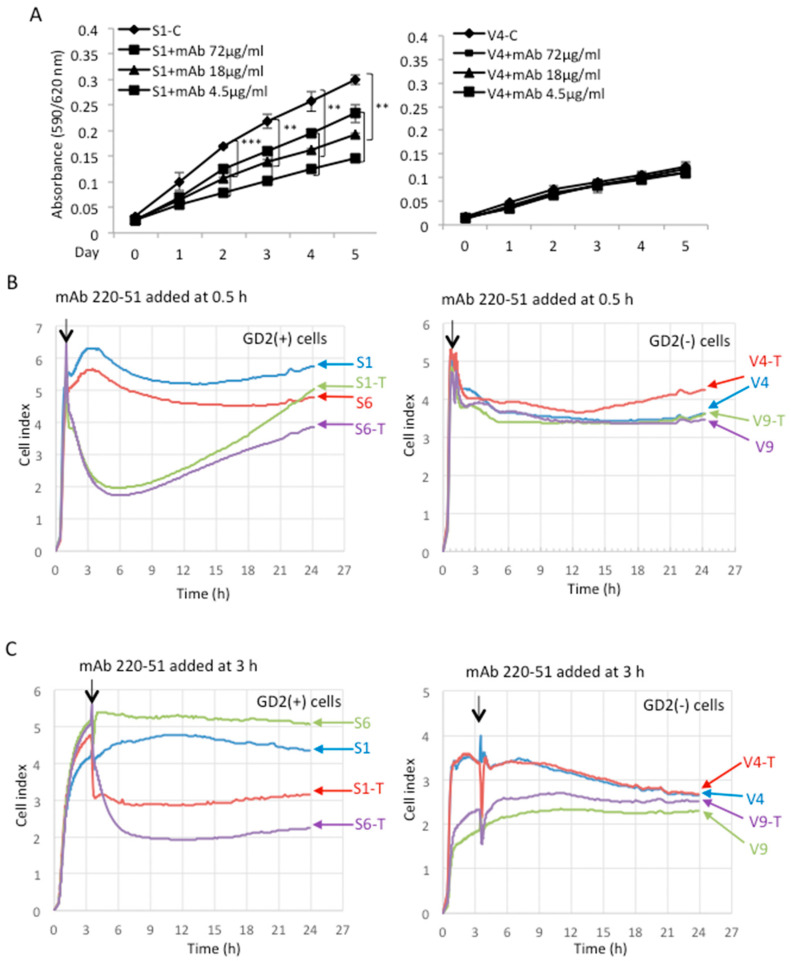
Effects of anti-GD2 mAbs on cell proliferation and adhesion of GD2+ cells. (**A**) Proliferation was analyzed by the MTT assay using GD2+ S1 and GD2− V4 cells. Cells were seeded in each well, and purified anti-GD2 mAbs 220-51 were added at 72, 18, and 4.5 μg/mL. The MTT solution was added and incubated on each day as indicated. Absorbance was measured at 590/620 nm. The experiment was performed in triplicates (and the mean ± SD are presented) and analyzed by two-way ANOVA with Tukey post-hoc test between control (S1-C or V4-C) vs. anti-GD2 treated cell lines (S1+mAbs or V4+mAbs). *** *p* < 0.005 and ** *p* < 0.01. (**B**,**C**) Effects of treatment with anti-GD2 mAb 220-51 on cell adhesion were analyzed by the RT-CES system, using GD2+ and GD2− cells in collagen I-precoated microplates. Cells were seeded in the plates containing 200 μL of culture medium. Purified anti-GD2 antibody (220-51), 72 μg/200 μL, was added to the wells at 0.5 h (**B**) and 3.0 h (**C**), after starting. Antibody-treated cells are indicated as ‘T’.

**Figure 4 ijms-23-00423-f004:**
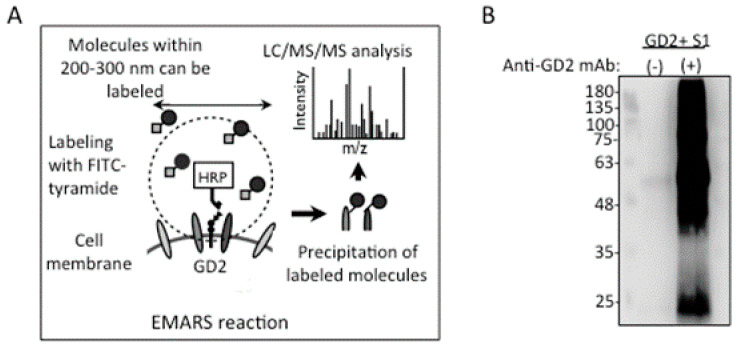
Identification of glycolipid-associated molecules by EMARS/MS. (**A**) A scheme of the EMARS/MS approach. EMARS was performed using GD2+ S1 cells, anti-GD2 mAb 220-51, and HRP-conjugated anti-mouse IgG. (**B**) Western blotting of EMARS reaction products. FITC-labeled molecules were immunoprecipitated and subsequently immunoblotted with anti-FITC antibodies. Bands were visualized using ImmunoStar^TM^ LD detection kits and analyzed by Imager LI-COR^TM^.

**Figure 5 ijms-23-00423-f005:**
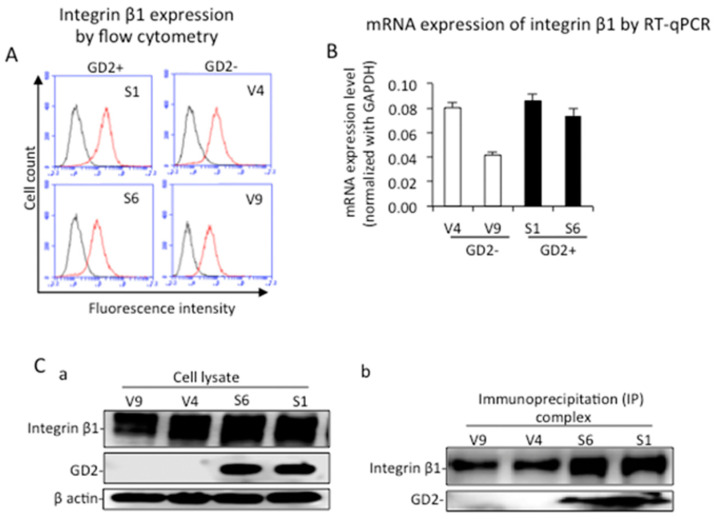
Physical association of GD2 and integrin β1. (**A**) Cell surface expressions of integrin β1 in GD2+ and GD2− cells were analyzed by flow cytometry using anti-integrin β1 mAb. (**B**) The mRNA expression of integrin β1 was analyzed by qRT-PCR using GD2+ and GD2− cells. The experiment was performed in triplicates and the mean ± SD are presented. (**Ca**) Expressions of integrin β1 as well as GD2 were analyzed by Western immunoblotting. GD2 and integrin β1 were detected separately with anti-GD2 mAb and anti-integrin β1 mAb. (**Cb**) The binding of ganglioside GD2 and integrin β1 was analyzed by immunoprecipitation with rabbit anti-integrin β1 antibodies, and subsequent immunoblotting with anti-integrin β1 mAb or anti-GD2 mAb.

**Figure 6 ijms-23-00423-f006:**
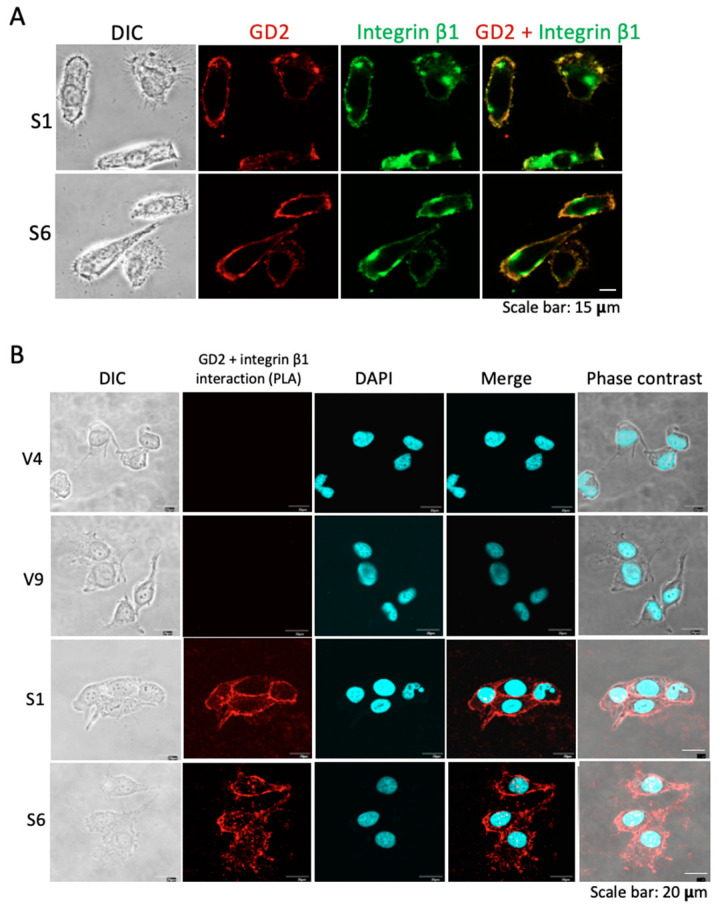
Colocalization of GD2 and integrin β1. (**A**) Cell surface localization of GD2 and integrin β1 was examined by immunocytochemistry using GD2+ cells. After fixation and permeabilization, cells were stained with mouse anti-GD2 mAb and rabbit anti-integrin β1 mAb. Then, cells were incubated with an Alexa 568-conjugated goat anti-mouse IgG antibody and an Alexa 488-conjugated donkey anti-rabbit IgG antibody. Microscopic visualization was performed using a confocal microscope. The green color indicates integrin β1 and the red color indicates GD2. Scale bar = 15 μm. (**B**) Association between GD2 and integrin β1 was analyzed by PLA. GD2+ and GD2− cells were incubated with anti-GD2 and anti-integrin β1 mAb. Duolink^TM^ in situ PLA probes anti-mouse PLUS and anti-rabbit MINUS were added, then a ligation–ligase solution was added. Finally, an amplification reaction was carried out. Cells were visualized under a confocal microscope. Scale bar = 20 μm.

**Figure 7 ijms-23-00423-f007:**
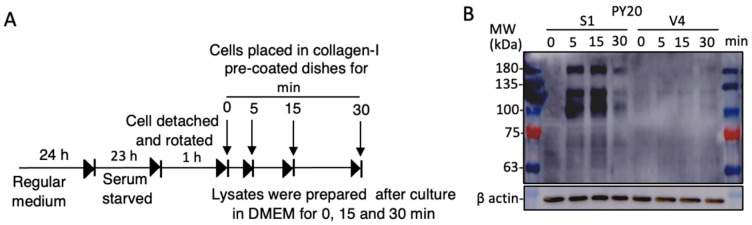
Adhesion signals activated during cell adhesion to collagen-coated plates. Tyrosine-phosphorylated proteins were detected during cell adhesion. (**A**) Diagram for preparing cells to obtain lysates during cell adhesion. GD2+ (S1) and GD2− (V4) cells were detached using 0.02% EDTA/PBS after culturing in serum-free DMEM at 37 °C for 23 h, and rotated at 37 °C for 1 h. Cell suspensions were placed in collagen I pre-coated plates in DMEM, and incubated as indicated at 37 °C. Then, cells were lysed and lysates were used for SDS-PAGE. (**B**) Immunoblotting was performed with anti-phosphotyrosine PY20 mAb. Anti-β actin Ab was used as a loading control.

**Figure 8 ijms-23-00423-f008:**
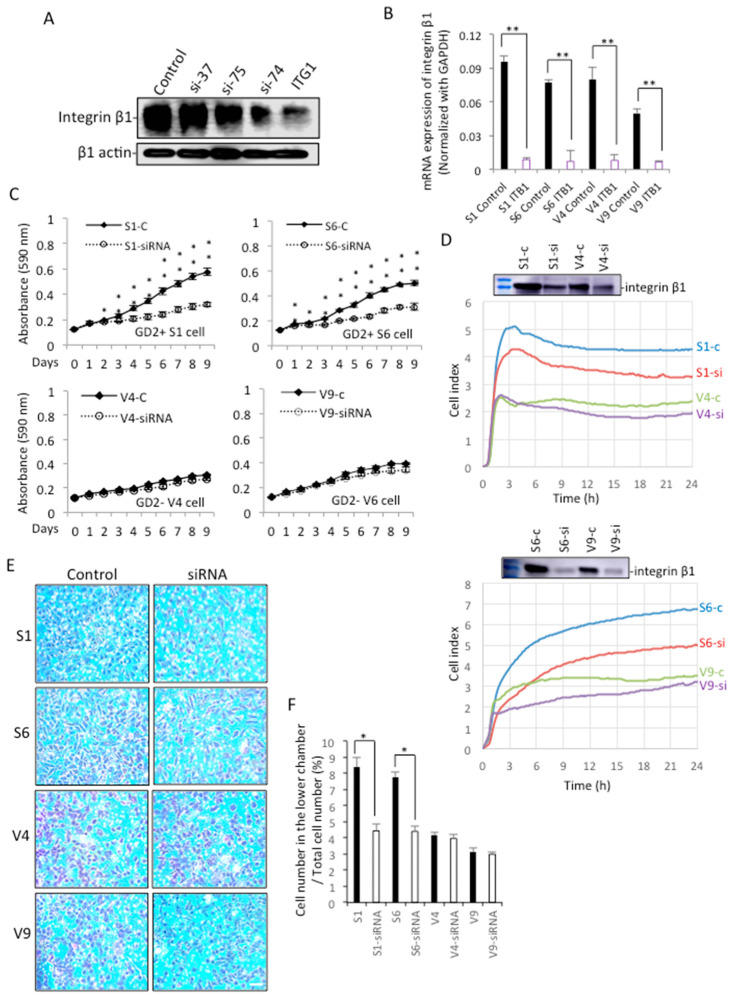
Knockdown of integrin β1 and its effects on cell phenotypes. (**A**,**B**) Knockdown efficiency of integrin β1 was examined with 4 types of siRNA (37, 75, 74, and ITG1). Using cell lysates and RNAs from GD2+ and GD2− cells, Western immunoblotting (**A**) and qRT-PCR (**B**) were performed, respectively. Gene expression levels were analyzed using the Student’s *t*-test. ** *p* < 0.01. (**C**) Cell proliferation was analyzed by the MTT assay, using GD2+ and GD2− cells treated with anti-integrin β1 si-RNA ITG1. Cells (3 × 10^3^) were seeded in 96-well plates. MTT assay was performed, as described in Figure 2. The analysis was performed in triplicates (and the mean ± SD are presented) and analyzed by two-way ANOVA with the Tukey post-hoc test. * *p* < 0.05, ** *p* < 0.01. (**D**) Cell adhesion was analyzed by the RT-CES system. GD2+ and GD2− cells were transfected with integrin β1 si-RNA, ITG1, and used for RT-CES, as described in Figure 2. (**E**,**F**) Invasion activity was analyzed using GD2+ and GD2− cells treated by integrin β1 si-RNA ITG1 with cell culture inserts. (**F**) A summary of the invasion assay. The invasion assay was performed in triplicates (and the mean ± SD are presented) were analyzed by Student’s *t*-test. * *p* < 0.05. Scale bar = 20 μm.

**Figure 9 ijms-23-00423-f009:**
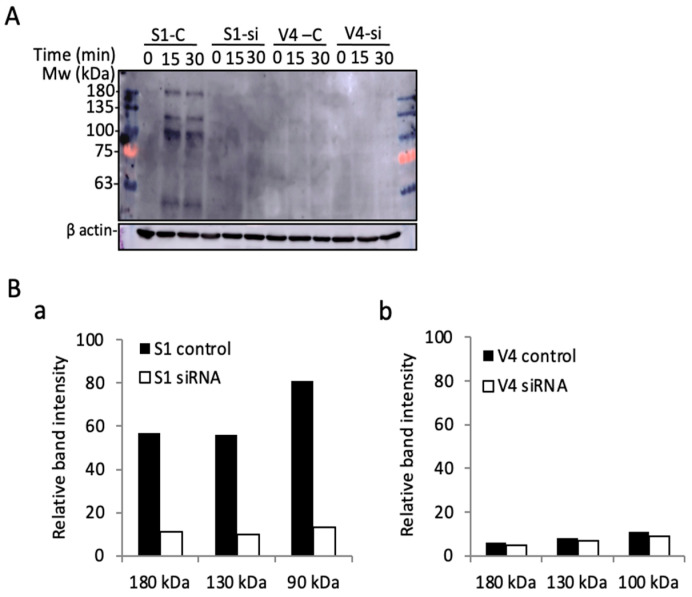
Multiple phospho-tyrosine bands were detected during the cell adhesion of GD2+ cells. (**A**) Immunoblotting with anti-phosphotyrosine mAb PY20. GD2+ S1 and GD2− V4 cells were transfected with anti-integrin β1 si-RNA ITG1. After 36 h of culture in regular medium, the cells were prepared, as described in Figure 7. Then, the cells were lysed and used for immunoblotting. (**B**) Band intensities in A were scanned by Images J^TM^ and plotted for S1 bands (**a**) and V4 bands (**b**).

**Figure 10 ijms-23-00423-f010:**
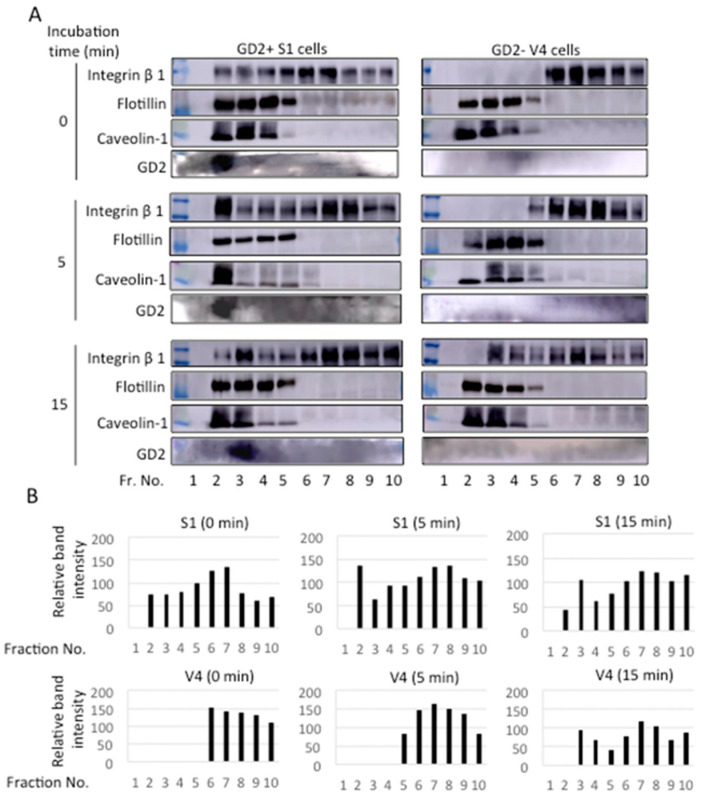
Intracellular distribution of integrin β1 before and during adhesion to CL type I. (**A**) GD2+ (S1) and control GD2− (V4) cells were detached using 0.5 mM EDTA/PBS. Cell suspension (5 × 10^5^ cells) was placed in collagen I pre-coated plates in DMEM, and incubated for 0~30 min at 37 °C. Then, cell lysates were prepared using 1% Brij 020 in a TNE buffer and separated by Optiprep gradient ultracentrifugation at 42,000 rpm and 4 °C for 5 h, and fractionated (500 μL). Each fraction (13.5 μL) was used for immunoblotting with anti-integrin β1 mAb, anti-GD2 mAb, anti-flotillin, or anti-caveolin-1 antibodies. Flotillin and caveolin-1 were used as GEM/raft markers. (**B**) Band intensities of integrin β1 were measured by ImageJ^TM^ software and the relative intensity of bands are presented.

## Data Availability

Not applied.

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
