# Peer review of "Ganglioside GD2 Enhances the Malignant Phenotypes of Melanoma Cells by Cooperating with Integrins"

_ijms, 2021, doi:10.3390/ijms23010423_

Round 1

Reviewer 1 Report

In this manuscript titled “Ganglioside GD2 enhances malignant phenotypes of melanoma cells by co-operating with integrins“ the authors have searched for GD2-associated molecules on the cell membrane using enzyme-mediated activation of radical sources combined with mass spectrometry, and identified integrin b1 as a representative GD2-associating molecule. In addition, they represented the physical association of GD2 and integrin b1 and intracellular distribution of integrins during cell adhesion to be localized in GEM/raft fractions in GD2+ cells, while localized in non-raft fractions in GD2- cells. The results suggest that GD2 and integrin b1 cooperate in GEM/rafts which could contribute to enhanced malignant phenotypes of melanomas.

This research contributes to increasing knowledge on GD2 ganglioside distribution in the cell membrane/membrane lipid rafts related to their function associated to other cell membrane molecules and cell signaling. The methodology and the results are very extensive and almost impressive, but missing adequate explanation and discussion regarding the results. The introduction is also very modest, the authors could introduce this interesting topic in more profound way and point out the importance of the study. Also, the EMARS/MS approach should have been explained in the introduction, with underlining the importance of the method and explaining the principle. The principle is, on the other hand, explained in the Figure 4, in the Results section, which is unusual, and this is the way most of the Figures are explained in the manuscript, with extensive methodology in Figure captions. I am not sure if this is completely not acceptable, I would consider it upon authors argumentation. Nevertheless, the methodology is already very extensive throughout the whole manuscript so I doubt in necessity of this in Figure captions.

The authors have generally succeeded to show the results of GD2 distribution and integrin-related phenotype of melanoma cells, but the manuscript leaves several important open questions;

  1. How did the authors choose integrin beta1 between 13 other molecules also detected in mAb-treated cells? The MS characterization results are shown in Supplement but only as a table with +/- of detection in mAB-treated cells. This characterization of integrin needs more clarification since the whole paper relies on the integrin as the only molecule detected in these cells. Is it possible to see raw MS data regarding this characterization?
  2. The statistical analysis is not argued nor explained. The statistical analysis of the extensive methodology is interpreted by only one sentence in section 4.16. The mean values have been compared, it is not stated the mean values of what precisely. The type of the distribution was not stated and what justification the authors had for using this particular test. Was it used for all experiments and why? Were the experiments done in parallel, in triplicates? This questions are important and left unanswered.
  3. The determination of cell signals in GD2+ cells, the authors claim that the identification of upper two bands from immunoblot (Figure 7.B) was done by MS, revealing them to be EGFR and 221 FAK, but no MS data of these results were represented in the manuscript. Is it possible to see these MS results?

The paper also needs extensive editing of English language and especially style, I have noted in the manuscript in several places unusual English formulations and not well articulated statements.

The manuscript has a potential due to valuable results, but the introduction, discussion, pointing out the importance of the research and results is not emphasized enough and not presented in well-structured manner.

There is considerable amount of self-citations.

The references are not uniformly written.

Author Response

Responses to the individual questions:

Reviewer 1.

His/her summary of the contents:

This research contributes to increasing knowledge on GD2 ganglioside distribution in the cell membrane/membrane lipid rafts related to their function associated to other cell membrane molecules and cell signaling. The methodology and the results are very extensive and almost impressive, but missing adequate explanation and discussion regarding the results. The introduction is also very modest, the authors could introduce this interesting topic in more profound way and point out the importance of the study. Also, the EMARS/MS approach should have been explained in the introduction, with underlining the importance of the method and explaining the principle. The principle is, on the other hand, explained in the Figure 4, in the Results section, which is unusual, and this is the way most of the Figures are explained in the manuscript, with extensive methodology in Figure captions. I am not sure if this is completely not acceptable, I would consider it upon authors argumentation. Nevertheless, the methodology is already very extensive throughout the whole manuscript so I doubt in necessity of this in Figure captions.

The authors have generally succeeded to show the results of GD2 distribution and integrin-related phenotype of melanoma cells, but the manuscript leaves several important open questions;

  1. How did the authors choose integrin beta1 between 13 other molecules also detected in mAb-treated cells? The MS characterization results are shown in Supplement but only as a table with +/- of detection in mAB-treated cells. This characterization of integrin needs more clarification since the whole paper relies on the integrin as the only molecule detected in these cells. Is it possible to see raw MS data regarding this characterization?

Ans: As the reviewer suggested to show raw MS data defining integrin b1, we have added related MS data in the Supplementary Fig. 3. Detailed procedures for LC-MS have also been described in Supplementary information. We have added a new sentence “Among 13 molecules defined, only integrin b1 was a definite membrane molecule” in 2,4.

  1. The statistical analysis is not argued nor explained. The statistical analysis of the extensive methodology is interpreted by only one sentence in section 4.16. The mean values have been compared, it is not stated the mean values of what precisely. The type of the distribution was not stated and what justification the authors had for using this particular test. Was it used for all experiments and why? Were the experiments done in parallel, in triplicates? This questions are important and left unanswered.

Ans: As the reviewer indicated, there were no explanation for the statistical evaluation of data in each figure. So, we added explanation for sample numbers and meaning of graphs etc in Figures 2, 3, 5, and 8. Namely, many of them were performed in triplicates, and mean ± S.D. were presented.

  1. The determination of cell signals in GD2+ cells, the authors claim that the identification of upper two bands from immunoblot (Figure 7.B) was done by MS, revealing them to be EGFR and FAK, but no MS data of these results were represented in the manuscript. Is it possible to see these MS results?

Ans: As the reviewer commented, we did not present details of MS analysis, since they were done by a standard method. But, we agree that MS results should increase the reliability of our data. So, we have added raw data of MS analysis to identify EGFR and FAK in Supplementary Fig. 4.

The paper also needs extensive editing of English language and especially style, I have noted in the manuscript in several places unusual English formulations and not well articulated statements.

Ans: This manuscript was checked by a native English speaker who is a specialist for scientific articles. But, we have tried to correct apparently peculiar places as much as we can.

The manuscript has a potential due to valuable results, but the introduction, discussion, pointing out the importance of the research and results is not emphasized enough and not presented in well-structured manner.

Ans: As the reviewer suggested, we have modified Introduction, and Discussion in order to emphasize 1. merit and principle of EMARS/MS approach, and 2. Importance of ganglioside GD2 by citing new and broad references.

There is considerable amount of self-citations.

Ans: As the reviewer indicated, there were many papers of our group cited. This is because this paper became a comprehensive dissertation covering various profiles of ganglioside function. So, we needed to cite many past papers of us, but we also cited many other papers related to the individual topics particularly in the revised version.

The references are not uniformly written.

Ans: As the reviewer commented, there were many not uniform descriptions in references. We have corrected them as much as we can.

Reviewer 2 Report

In the study, Yesmin et al. demonstrated that ganglioside GD2 altered the phenotype of melanoma cells by binding to integrin-β1. The ganglioside GD2 overexpressing melanoma cell line was used to reveal that GD2 increased proliferation, invasion, and adhesion. Moreover, the knockout of integrin-β1 cancelled the effects derived from ganglioside GD2. The authors used different methods to prove the cooperation of ganglioside GD2 and integrin-β1, which makes the evidence solid. However, there some points could be improved.

  1. The authors have to provide more information why ganglioside GD2 was selected and the importance of ganglioside GD2 in melanoma in the introduction.
  2. Is The expression level of ganglioside GD2 higher in the cancer cells than normal cells?
  3. Other melanoma cell lines should be used to investigate whether the effect is general.
  4. Metastasis was usually associated with increase of migration and invasion and decrease of adhesion. Moreover, the cells with high level of invasion often have poor adhesion ability. However, the authors concluded that ganglioside GD2 enhanced metastasis because both invasion and adhesion were induced in GD2+ melanoma cells. Thus, the mechanism of the abnormal situation must be explained.
  5. The authors included too many descriptions of method in the results and figure legends, which makes it very difficult to read and follow their idea.
  6. What was the baseline levels of integrin-β1 in Fig. 10b for calculating relative intensity?

Author Response

Responses to the individual questions:

Reviewer 2.

In the study, Yesmin et al. demonstrated that ganglioside GD2 altered the phenotype of melanoma cells by binding to integrin-β1. The ganglioside GD2 overexpressing melanoma cell line was used to reveal that GD2 increased proliferation, invasion, and adhesion. Moreover, the knockout of integrin-β1 cancelled the effects derived from ganglioside GD2. The authors used different methods to prove the cooperation of ganglioside GD2 and integrin-β1, which makes the evidence solid. However, there some points could be improved.

  1. The authors have to provide more information why ganglioside GD2 was selected and the importance of ganglioside GD2 in melanoma in the introduction.

Ans: As the reviewer suggested, we have added many explanations on the importance of ganglioside GD2 in melanomas and also in other cancers in Introduction by citing several new references.

  1. Is The expression level of ganglioside GD2 higher in the cancer cells than normal cells?

Ans: Of course, ganglioside GD2 is expressed almost exclusively in cancer cells except restricted sites of nervous system. That is why GD2 is called as cancer-associated glycolipids. This point was described in the first sentence of Introduction, and in Discussion also.

  1. Other melanoma cell lines should be used to investigate whether the effect is general.

Ans: As the reviewer commented, we also feel that it is needed to confirm these results using other melanoma cell lines, although we wrote this manuscript believing that these results showed general facts. To draw the conclusion presented here, we have taken many steps starting from isolation of a ganglioside-deficient mutant subline of melanoma, isolation of cDNAs for ganglioside synthesis, to the establishment of transfectant cells to remodel ganglioside expression pattern, etc. Thus, it took so may years and a lot of efforts to achieve these results only for one melanoma line. Therefore, it seems too much to include similar data using other melanoma lines. We are sure to try it in the near future.

  1. Metastasis was usually associated with increase of migration and invasion and decrease of adhesion. Moreover, the cells with high level of invasion often have poor adhesion ability. However, the authors concluded that ganglioside GD2 enhanced metastasis because both invasion and adhesion were induced in GD2+ melanoma cells. Thus, the mechanism of the abnormal situation must be explained.

Ans: As the reviewer indicated, increased adhesion and increased invasion activity frequently appear to co-exist in malignant cells depending on the assay systems. These results may propose a contradictory interpretation. We are now thinking this is due to conditions of the individual assay systems, and essentially due to signaling molecular profiles working in the examined cells. Especially for invasion activity, obtained results are sometimes opposite depending on the assay conditions, because this assay includes multiple biological processes such as digestion of extracellular matrix, cell motility etc. We think it should be important to clearly define the assay conditions under which some results are obtained. Therefore, we carefully described conditions of our invasion assay in Materials and Methods, and in Results. Furthermore, we explained about those results in Discussion on p12, lines 1~18 from the bottom (344-351) by comparing results in this paper and those in the past report.

  1. The authors included too many descriptions of method in the results and figure legends, which makes it very difficult to read and follow their idea.

Ans: As the reviewer indicated, we described similar explanation about methods in Results and figure legends, disturbing readers’ understanding. So, we have made it more concise in Results and Figure legends, leaving solid descriptions in Materials and Methods as they are. Especially for Figure 4, explanation for EMARS/MS has largely been shortened.

  1. What was the baseline levels of integrin-β1 in Fig. 10b for calculating relative intensity?

Ans: We used intensity of the first lane in the measurement of band intensities by ImageJ, since this fraction usually contained no significant components. We believe that this is comparison of band intensity of particular protein only among fractions under particular condition.

Round 2

Reviewer 1 Report

The authors have only answered to one major question regarding the identification of upper two bands from immunoblot (Figure 7.B) that was done by MS, revealing them to be EGFR and FAK and provided the MS data to clarify this. The authors have also added some citations in the Introduction and Discussion in order to emphasize the principle of EMARS/MS approach and importance of ganglioside GD2.

Nevertheless, they have not answered nor provided the MS data for the major questions regarding identification of integrin and statistical analysis:

  1. Comment from the 1st Revision regarding integrin:

„How did the authors choose integrin beta1 between 13 other molecules also detected in mAb-treated cells? The MS characterization results are shown in Supplement but only as a table with +/- of detection in mAB-treated cells. This characterization of integrin needs more clarification since the whole paper relies on the integrin as the only molecule detected in these cells. Is it possible to see raw MS data regarding this characterization?“

The authors have provided in Supplementary file just figures for MS characterization of EGFR and FAK, but not for integrin beta, which was the most important issue, as I have pointed out in my comment in the first review that the whole paper relies on this.

  1. Comment from the 1st Revision regarding statistical analysis:
  1. „The statistical analysis is not argued nor explained. The statistical analysis of the extensive methodology is interpreted by only one sentence in section 4.16. The mean values have been compared, it is not stated the mean values of what precisely. The type of the distribution was not stated and what justification the authors had for using this particular test. Was it used for all experiments and why? Were the experiments done in parallel, in triplicates? This questions are important and left unanswered.“

The authors have just added in the captions of Figures 2, 3 and 5 that the analysis were made in triplicates, while the section 4.16. Statistical analysis has not been modified nor changed.

Therefore I still do not recommend this manuscript for publishing in IJMS in this form since the major questions have not been answered nor explained.

Best regards

Author Response

Reviewer 1:

Summary of the revision.

Nevertheless, they have not answered nor provided the MS data for the major questions regarding identification of integrin and statistical analysis:

  1. Comment from the 1stRevision regarding integrin:

How did the authors choose integrin beta1 between 13 other molecules also detected in mAb-treated cells? The MS characterization results are shown in Supplement but only as a table with +/- of detection in mAB-treated cells. This characterization of integrin needs more clarification since the whole paper relies on the integrin as the only molecule detected in these cells. Is it possible to see raw MS data regarding this characterization?“

The authors have provided in Supplementary file just figures for MS characterization of EGFR and FAK, but not for integrin beta, which was the most important issue, as I have pointed out in my comment in the first review that the whole paper relies on this.

Ans: As the reviewer indicated, MS data for the identification of integrin b only in mAb+ sample was missing by accident. We feel so sorry. We have newly added Supplementary Fig. 3 ~ 4 including MS data of integrin beta.

  1. Comment from the 1stRevision regarding statistical analysis:

The statistical analysis is not argued nor explained. The statistical analysis of the extensive methodology is interpreted by only one sentence in section 4.16. The mean values have been compared, it is not stated the mean values of what precisely. The type of the distribution was not stated and what justification the authors had for using this particular test. Was it used for all experiments and why? Were the experiments done in parallel, in triplicates? This questions are important and left unanswered.“

The authors have just added in the captions of Figures 2, 3 and 5 that the analysis were made in triplicates, while the section 4.16. Statistical analysis has not been modified nor changed.

Therefore I still do not recommend this manuscript for publishing in IJMS in this form since the major questions have not been answered nor explained.

Ans: As the reviewer commented, the explanation for statistical analysis was not sufficient and too simple. We have re-analyzed results in Fig. 2A, 3A and 8C, and explained details about methods used in the individual legends. Accordingly, we have modified graphs in these figures. We have also added explanations for the statistical significance for other data also. As for 4.16, we have modified and enriched the description on the statistical analysis covering all data analyses.

Reviewer 2 Report

The authors addressed all the comments. There is no further question.

Author Response

Reviewer 2:

 Comments and Suggestions for Authors

The authors addressed all the comments. There is no further question.

Ans: Thank you very much. We are so pleased to hear the reviewer’s decision. Although we have slightly modified text and figures, we believe that these changes do not disturb his/her decision.

Round 3

Reviewer 1 Report

I would recomend the manuscript for publishing after the authors have answered the most important issues I have adressed in my reviews in their second revision. There are still some spelling corrections neccessary, and Image J is written with capital "I".